# Leveraging Simple Model Predictions for Enhancing its Performance

## Abstract

There has been recent interest in improving performance of simple models for multiple reasons such as interpretability, robust learning from small data, deployment in memory constrained settings as well as environmental considerations. In this paper, we propose a novel method SRatio that can utilize information from high performing complex models (viz. deep neural networks, boosted trees, random forests) to reweight a training dataset for a potentially low performing simple model such as a decision tree or a shallow network enhancing its performance. Our method also leverages the per sample hardness estimate of the simple model which is not the case with the prior works which primarily consider the complex model's confidences/predictions and is thus conceptually novel. Moreover, we generalize and formalize the concept of attaching probes to intermediate layers of a neural network to other commonly used classifiers and incorporate this into our method. The benefit of these contributions is witnessed in the experiments where on 6 UCI datasets and CIFAR-10 we outperform competitors in a majority (16 out of 27) of the cases and tie for best performance in the remaining cases. In fact, in a couple of cases, we even approach the complex model's performance. We also conduct further experiments to validate assertions and intuitively understand why our method works. Theoretically, we motivate our approach by showing that the weighted loss minimized by simple models using our weighting upper bounds the loss of the complex model.

## 1 Introduction

Simple models such as decision trees or rule lists or shallow neural networks still find use in multiple settings where a) (global) interpretability is needed, b) small data sizes are available, or c) memory/computational constraints are prevalent (Dhurandhar et al., 2018b). In such settings compact or understandable models are often preferred over high performing complex models, where the combination of a human with an interpretable model can have better on-field performance than simply using the best performing black box model (Varshney et al., 2018). For example, a manufacturing engineer with an interpretable model may be able to obtain precise knowledge of how an out-of-spec product was produced and can potentially go back to fix the process as opposed to having little-to-no knowledge of how the decision was reached. Posthoc local explainability methods (Ribeiro et al., 2016; Bach et al., 2015; Dhurandhar et al., 2018a) can help delve into the local behavior of black box models, however, besides the explanations being only local, there is no guarantee that they are in fact true (Rudin, 2018). There is also a growing concern of the carbon footprint left behind in training complex deep models (Strubell et al., 2019), which for some popular architectures is more than that left behind by a car over its entire lifetime.

In this paper, we propose a method, SRatio, which reweights the training set to improve simple models given access to a highly accurate complex model such as a deep neural network, boosted trees, or some other predictive model. Given the applications we are interested in, such as interpretability or deployment of models in resource limited settings, we assume the complexity of the simple models to be predetermined or fixed (viz. decision tree of height $\leq 5$). We cannot grow arbitrary size ensembles such as in boosting or bagging (Freund & Schapire, 1997). Our method applies potentially to any complex-simple model combination which is not the case for some state-of-the-art methods in this space such as Knowledge Distillation (Geoffrey Hinton, 2015) or Profweight (Dhurandhar et al.,

2018b), where the complex model is assumed to be a deep neural network. In addition, we generalize and formalize the concept of probes presented in (Dhurandhar et al., 2018b) and provide examples of what they would correspond to for classifiers other than neural networks. Our method also uses the a priori low performing simple model's confidences to enhance its performance. We believe this to be conceptually novel compared to existing methods which seem to *only* leverage the complex model (viz. its predictions/confidences). The benefit is seen in experiments where we outperform other competitors in a majority of the cases and are tied with one or more methods for best performance in the remaining cases. In fact, in a couple of cases we even approach the complex model's performance, i.e. a single tree is made to be as accurate as 100 boosted trees. Moreover, we motivate our approach by contrasting it with covariate shift and show that our weighting scheme where we now minimize the weighted loss of the simple model is equivalent to minimizing an upper bound on the loss of the complex model.

## 2 RELATED WORK

Knowledge Distillation (Geoffrey Hinton, 2015; Tan et al., 2017; Lopez-Paz et al., 2016) is one of the most popular approaches for building "simpler" neural networks. It typically involves minimizing the cross-entropy loss of a simpler network based on calibrated confidences (Guo et al., 2017) of a more complex network. The simpler networks are usually not that simple in that they are typically of the same (or similar) depth but thinned down (Romero et al., 2015). This is generally insufficient to meet tight resource constraints (Reagen et al., 2016). Moreover, the thinning down was shown for convolutional neural networks but it is unclear how one would do the same for modern architectures such as ResNets. The weighting of training inputs approach on the other hand can be more easily applied to different architectures. It also has another advantage in that it can be readily applied to models optimizing losses other than cross-entropy (viz. hinge loss, squared loss) with some interpretation of which inputs are more (or less) important. Some other strategies to improve simple models (Buciluǎ et al., 2006; Ba & Caurana, 2013; Bastani et al., 2017) are also conceptually similar to Distillation, where the actual outputs are replaced by predictions from the complex model.

In Dehghani et al. (2017), authors use soft targets and their uncertainty estimates to inform a student model on a larger dataset with more noisy labels. Uncertainty estimates are obtained from Gaussian Process Regression done on a dataset that has less noisy labels. In Furlanello et al. (2018), authors train a student neural network that is identically parameterized to the original one by fitting to soft scores rescaled by temperature. In our problem, the complexity of the student is very different from that of the teacher and we do compare with distillation-like schemes. Frosst & Hinton (2017) define a new class of decision trees called soft decision trees to enable it to fit soft targets (classic distillation) of a neural network. Our methods use existing training algorithms for well-known simple models. Ren et al. (2018) advocate reweighting samples as a way to make deep learning robust by tuning the weights on a validation set through gradient descent. Our problem is about using knowledge from a pre-trained complex model to improve a simple model through weighting samples.

The most relevant work to our current endeavor is ProfWeight (Dhurandhar et al., 2018b), where they too weight the training inputs. Their method however, requires the complex model to be a neural network and thus does not apply to settings where we have a different complex model. Moreover, their method, like Distillation, takes into account only the complex model's assessment of an example's difficulty.

Curriculum learning (CL) (Bengio et al., 2009) and boosting (Freund & Schapire, 1997) are two other approaches which rely on weighting samples, however, their motivation and setup are significantly different. In both CL and boosting the complexity of the improved learner can increase as they do not have to respect constraints such as interpretability (Dhurandhar et al., 2017; Montavon et al., 2017) or limited memory/power (Reagen et al., 2016; Chen et al., 2016). In CL, typically, there is no automatic gradation of example difficulty during training. In boosting, the examples are graded with respect to a previous weak learner and not an independent accurate complex model. Also, as we later show, our method does not necessarily up-weight hard examples but rather uses a measure that takes into account hardness as assessed by both the complex and simple models.

## 3 METHODOLOGY

In this section, we first provide theoretical and intuitive justification for our approach. This is followed by a description of our method for improving simple models using both the complex and simple model's predictions. The key novelty lies in the use of the simple model's prediction, which also makes the theory non-trivial yet practical. Rather than use the complex model's prediction, we generalize a concept from (Dhurandhar et al., 2018b) where they attached probe functions to each layer of a neural network, obtained predictions using only the first $k$ layers ($k$ being varied up to the total number of layers), and used the mean of the probe predictions rather than the output of only the last layer. They empirically showed the extra information from previous layers to improve upon only using the final layer's output. Our generalization, which we call graded classifiers and formally define below, extracts progressive information from other models (beyond neural networks). The graded classifiers provide better performance than using only the output of complex model, as illustrated by our various experiments in the subsequent section.

### 3.1 THEORETICAL MOTIVATION

Our approach in section 3.3 can be motivated by contrasting it with the covariate shift (Agarwal et al., 2011) setting. If $X \times Y$ is the input-output space and $p(x, y)$ and $q(x, y)$ are the source and target distributions in the covariate shift setting, then it is assumed that $p(y|x) = q(y|x)$ but $p(x) \neq q(x)$. One of the standard solutions for such settings is importance sampling where the source data is sampled proportional to $\frac{q(x)}{p(x)}$ in order to mimic as closely as possible the target distribution. In our case, the dataset is the same but the classifiers (i.e. complex and simple) are different. We can think of this as a setting where $p(x) = q(x)$ as both the models learn from the same data, however, $p(y|x) \neq q(y|x)$ where $p(y|x)$ and $q(y|x)$ correspond to the outputs of complex and simple classifiers, respectively. Given that we want the simple model to approach the complex models performance, a natural analog to the importance weights used in covariate shift is to weight samples by $\frac{p(y|x)}{q(y|x)}$ which is the essence of our method as described below in section 3.3.

Now, let us formally show that the expected cross-entropy loss of a model is no greater than the reweighted version with an additional positive slack term. This implies that training the simple model with this reweighting is a valid and sound procedure of the loss we want to optimize.

**Lemma 3.1.** *Let $p_\theta(y|x)$ be the softmax scores on a specific model $\theta$ from simple model space $\Theta$. Let $\theta^* \in \Theta$ be the set of simple model parameters that is obtained from a given learning algorithm for the simple model on a training dataset. Let $p_c(y|x)$ be a pre-trained complex classifier whose loss is smaller than $\theta^*$ on the training distribution. Let $\beta \geq 1$ be a scalar clip level for the ratio $p_c(y|x)/p_{\theta^*}(y|x)$. Then we have:*

$$\mathbb{E}[-\log p_\theta(y|x)] \leq \mathbb{E}\left[\max\left(1, \min\left(\frac{p_c(y|x)}{p_{\theta^*}(y|x)}, \beta\right)\right) \log\left(\frac{1}{p_\theta(y|x)}\right)\right]$$
$$- \mathbb{E}\left[\log\left(\min\left(\frac{p_c(y|x)}{p_{\theta^*}(y|x)}, \beta\right)\right)\right] + \log(\beta). \quad (1)$$

*Proof.*

$$\mathbb{E}[-\log(p_\theta(y|x))] = \mathbb{E}\left[\log\left(\frac{1}{p_\theta(y|x)} \cdot \min\left(\frac{p_c(y|x)}{p_{\theta^*}(y|x)}, \beta\right)\right)\right] - \mathbb{E}\left[\log\left(\min\left(\frac{p_c(y|x)}{p_{\theta^*}(y|x)}, \beta\right)\right)\right]$$
$$\leq \mathbb{E}\left[\log\left(\frac{1}{p_\theta(y|x)} \cdot \max\left(1, \min\left(\frac{p_c(y|x)}{p_{\theta^*}(y|x)}, \beta\right)\right)\right)\right]$$
$$- \mathbb{E}\left[\log\left(\min\left(\frac{p_c(y|x)}{p_{\theta^*}(y|x)}, \beta\right)\right)\right]$$
$$\overset{a}{\leq} \mathbb{E}\left[\max\left(1, \min\left(\frac{p_c(y|x)}{p_{\theta^*}(y|x)}, \beta\right)\right) \log\left(\frac{1}{p_\theta(y|x)}\right)\right]$$
$$- \mathbb{E}\left[\log\left(\min\left(\frac{p_c(y|x)}{p_{\theta^*}(y|x)}, \beta\right)\right)\right] + \log(\beta) \quad (2)$$

(a) The inequality $\log(wx) \leq w \log(x) + \log(\beta)$ holds for all $\beta \geq w \geq 1, x > 1$ where $\beta \geq 1$ is any arbitrary clip level.

$\square$

**Remark 1:** We observe that re-weighing every sample by $\max(1, \min(\frac{p_c(y|x)}{p_{\theta*}(y|x)}, \beta))$ and re-optimizing using the simple model training algorithm is a sound way to optimize the cross-entropy loss of the simple model on the training data set. The reason we believe that optimizing the upper bound could be better is because many simple models such as decision trees are trained using a simple greedy approach. Therefore, reweighting samples based on an accurate complex model could induce the appropriate bias leading to better solutions. Moreover, in equation 2, the second inequality is the main place where there is slack between the upper bound and the quantity we are interested in bounding. This inequality exhibits a smaller slack if $w = \frac{p_c(y|x)}{p_{\theta*}(y|x)}$ is not much smaller than 1 with high probability. This is typical when $p_c$ comes from a more complex model that is more accurate than that of $\theta^*$.

**Remark 2:** The upper bound used for the last inequality in the proof leads to a quantification of the bias introduced by weighting for a particular dataset. Note that in practice, we determine the optimal $\beta$ via cross-validation.

### 3.2 INTUITIVE JUSTIFICATION

Intuitively, assuming $w \geq 1$ implies that the complex model finds an input easier (i.e. higher score or confidence) to classify in the correct class than does the simple model. Although in practice this may not always be the case, it is not unreasonable to believe that this would occur for most inputs, especially if the complex model is highly accurate.

The motivation for our approach conceptually does not contradict (Dhurandhar et al., 2018b), where hard samples for a complex model are weighted low. These would still be potentially weighted low as the numerator would be small. However, the main difference would occur in the weighting of the easy examples for the complex model, which rather than being uniformly weighted high, would now be weighted based on the assessment of the simple model. This, we believe, is important information as stressing inputs that are already extremely easy for the simple model to classify will possibly not lead to the best generalization. It is probably more important to stress inputs that are somewhat hard for the simple model but easier for the complex model, as that is likely to be the critical point of information transfer. Even though easier inputs for the complex model are likely to get higher weights, ranking these based on the simple model's assessment is important and not captured in previous approaches. Hence, although our idea may appear to be simple, we believe it is a significant jump conceptually in that it also takes into account the simple model's behavior to improve itself.

Our method described in the next section is a generalization of this idea and the motivation presented in the previous section. If the confidences of the complex model are representative of difficulty then we could leverage them alone. However, many times as seen in previous work (Dhurandhar et al., 2018b), they may not be representative and hence using confidences of lower layers or simpler forms of the complex classifier can be very helpful.

### 3.3 METHOD

We now present our approach SRatio in algorithm 1, which uses the ideas presented in the previous sections and generalizes the method in (Dhurandhar et al., 2018b) to be applicable to complex classifiers other than neural networks.

In previous works (Akshayvarun Subramanya, 2017; Dhurandhar et al., 2018b), it was seen that sometimes highly accurate models such as deep neural networks may not be good density estimators and hence may not provide an accurate quantification of the relative difficulty of an input. To obtain a better quantification, the idea of attaching probes (viz. linear classifiers) to intermediate layers of a deep neural network and then averaging the confidences was proposed. This, as seen in the previous work, led to significantly better results over the state-of-the-art. Similarly, we generalize our method where rather than taking just the output confidences of the complex model as the numerator,

---

**Algorithm 1** Our proposed method SRatio.

**Input:** $n$ (graded) classifiers $\zeta_1, ..., \zeta_n$, learning algorithm for simple model $\mathcal{L}_\mathcal{S}$, dataset $D_\mathcal{S}$ of cardinality $N$, performance gap parameter $\gamma$ and maximum allowed ratio parameter $\beta$.

1) Train simple model on $D_\mathcal{S}$, $\mathcal{S} \leftarrow \mathcal{L}_\mathcal{S}(D_\mathcal{S}, \vec{1}_N)$ and compute its (average) prediction error $\epsilon_\mathcal{S}$.{Obtain initial simple model where each input is given a unit weight.}
2) Compute (average) prediction errors $\epsilon_1, ..., \epsilon_n$ for the $n$ graded classifiers and store the ones that are at least $\gamma$ more accurate than the simple model i.e. $I \leftarrow \{i \in \{1, ..., n\} \mid \epsilon_\mathcal{S} - \epsilon_i \geq \gamma\}$
3) Compute weights for all inputs $x$ as follows: $w(x) = \frac{\sum_{i \in I} \zeta_i(x)}{m\mathcal{S}(x)}$, where $m$ is the cardinality of set $I$ and $\mathcal{S}(x)$ is the prediction probability/score for the true class of the simple model.
4) Set $w(x) \leftarrow 0$, if $w(x) > \beta$. {Ignore extremely hard examples for the simple model.}
5) Retrain the simple model on the dataset $D_\mathcal{S}$ with the corresponding learned weights $\mathbf{w}$, $\mathcal{S}_w \leftarrow \mathcal{L}_\mathcal{S}(D_\mathcal{S}, \mathbf{w})$
6) **Return** $\mathcal{S}_w$

---

we take an average of the confidences over a gradation of outputs produced by taking appropriate simplifications of the complex model. We formalize this notion of graded outputs as follows:

**Definition ($\delta$-graded)** Let $X \times Y$ denote the input-output space and $p(x, y)$ the joint distribution over this space. Let $\zeta_1, \zeta_2, ..., \zeta_n$ denote classifiers that output the prediction probabilities for a given input $x \in X$ for the most probable (or true) class $y \in Y$ determined by $p(y|x)$. We then say that classifiers $\zeta_1, \zeta_2, ..., \zeta_n$ are $\delta$-graded for some $\delta \in (0, 1]$ and a (measurable) set $Z \subseteq X$ if $\forall x \in Z$, $\zeta_1(x) \leq \zeta_2(x) \leq \cdots \leq \zeta_n(x)$, where $\int_{x \in Z} p(x) \geq \delta$.

Loosely speaking, the above definition says that a sequence of classifiers is $\delta$-graded if a classifier in the sequence is at least as accurate as the ones preceding it for inputs whose probability measure is at least $\delta$. Thus, a sequence would be 1-graded if the above inequalities were true for the entire input space (i.e. $Z = X$). Below are some examples of how one could produce $\delta$-graded classifiers for different models in practice.

- Deep Neural Networks: The notion of attaching probes, which are essentially linear classifiers (viz. $\sigma(Wx + b)$) trained on intermediate layers of a deep neural network (Dhurandhar et al., 2018b; Alain & Bengio, 2016) could be seen as a way of creating $\delta$-graded classifiers, where lower layer probes are likely to be less accurate than those above them for most of the samples. Thus the idea of probes as simplifications of the complex model, as used in previous works, are captured by our definition.

- Boosted Trees: One natural way here could be to consider the ordering produced by boosting algorithms that grow the tree ensemble and use all trees up to a certain point. For example, if we have an ensemble of 10 trees, then $\zeta_1$ could be the first tree, $\zeta_2$ could be the first two trees and so on where $\zeta_{10}$ is the entire ensemble.

- Random Forests: Here one could order trees based on performance and then do a similar grouping as above where $\zeta_1$ could be the least accurate tree, then $\zeta_2$ could be the ensemble of $\zeta_1$ and the second most inaccurate tree and so on. Of course, for this and boosted trees one could take bigger steps and add more trees to produce the next $\zeta$ so that there is a measurable jump in performance from one graded classifier to the next.

- Other Models: For non-ensemble models such as generalized linear models one too could form graded classifiers by taking different order Taylor approximations of the functions, or by setting the least important coefficients successively to zero by doing function decompositions based on binary, ternary and higher order interactions (Molnar et al., 2019), or using feature selection and starting with a model containing the most important feature(s).

Given this, we see in algorithm 1 that we take as input graded classifiers and the learning algorithm for the simple model. Trivially, the graded classifiers can just be the entire complex classifier where we only consider its output confidences. We now take a ratio of the average confidence of the graded classifiers that are at least more accurate than the simple model by $\gamma > 0$ and the simple model's confidence. If this ratio is too large (i.e. $> \beta$) we set the weight to zero and otherwise the ratio is the

Table 1: Below we see the averaged % errors with 95% confidence intervals for the different methods on six real datasets. Boosted Trees and Random Forest (100 trees) are the complex models (CM), while a single decision tree and linear SVM are the simple models (SM). Best simple model results are indicated in bold. ∗ indicates the simple model has approached the complex models performance.

| Dataset | Complex Model | CM Error | Simple Model | SM Error | Distill-proxy 1 Error (SM) | ConfWeight Error (SM) | SRatio Error (SM) |
|---|---|---|---|---|---|---|---|
| Ionosphere | Boosted Trees | 8.10 ±0.4 | Tree | 10.95 ±0.4 | 10.95 ±0.4 | 11.42 ±0.8 | **8.57**∗ ±**0.5** |
| | | | SVM | 12.38 ±0.6 | 11.90 ±0.6 | 11.90 ±0.6 | **10.47** ±**0.5** |
| | Random Forest | 6.19 ±0.4 | Tree | 10.95 ±0.4 | 10.95 ±0.4 | 11.42 ±0.4 | **10.42** ±**0.1** |
| | | | SVM | 12.38 ±0.6 | 12.38 ±0.6 | 12.38 ±0.6 | **11.42** ±**0.3** |
| Ovarian Cancer | Boosted Trees | 4.68 ±0.4 | Tree | **15.62** ±**0.8** | **15.62** ±**0.8** | **15.62** ±**1.0** | **15.62** ±**0.5** |
| | | | SVM | **1.56** ±**0.4** | **1.56** ±**0.4** | **1.56** ±**0.4** | **1.56** ±**0.4** |
| | Random Forest | 6.25 ±0.8 | Tree | 15.62 ±0.8 | 15.62 ±0.8 | **14.06** ±**0.1** | **14.04** ±**0.1** |
| | | | SVM | **1.56** ±**0.4** | **1.56** ±**0.4** | **1.56** ±**0.4** | **1.56** ±**0.4** |
| Heart Disease | Boosted Trees | 15.55 ±0.6 | Tree | 23.88 ±0.7 | **22.77** ±**0.1** | 23.33 ±0.3 | **22.77** ±**0.2** |
| | | | SVM | 17.22 ±0.2 | **16.67** ±**0.3** | 17.22 ±0.2 | **16.77** ±**0.2** |
| | Random Forest | 15.88 ±0.6 | Tree | 23.88 ±0.7 | 23.88 ±0.7 | 25.55 ±0.5 | **22.77** ±**0.3** |
| | | | SVM | 17.22 ±0.2 | 17.22 ±0.2 | **16.67** ±**0.3** | **16.67** ±**0.2** |
| Waveform | Boosted Trees | 12.96 ±0.1 | Tree | 25.43 ±0.2 | **25.06** ±**0.1** | **25.10** ±**0.1** | **25.06** ±**0.1** |
| | | | SVM | 14.70 ±0.2 | 15.33 ±0.0 | 14.70 ±0.2 | **13.72** ±**0.2** |
| | Random Forest | 10.90 ±0.1 | Tree | 25.43 ±0.2 | 25.43 ±0.2 | 25.43 ±0.2 | **25.06** ±**0.1** |
| | | | SVM | 14.70 ±0.2 | 14.33 ±0.0 | 14.30 ±0.2 | **12.72** ±**0.5** |
| Human Activity Recognition | Boosted Trees | 6.32 ±0.0 | Tree | 7.93 ±0.2 | 7.93 ±0.1 | 7.86 ±0.2 | **7.15** ±**0.1** |
| | | | SVM | 14.56 ±0.1 | 15.85 ±0.1 | **13.92** ±**0.1** | **13.92** ±**0.2** |
| | Random Forest | 2.34 ±0.0 | Tree | 7.93 ±0.2 | 7.23 ±0.1 | 7.21 ±0.1 | **6.67** ±**0.0** |
| | | | SVM | 14.56 ±0.1 | **13.92** ±**0.1** | 14.24 ±0.1 | **13.92** ±**0.1** |
| Musk | Boosted Trees | 4.06 ±0.1 | Tree | 4.49 ±0.1 | 6.11 ±0.1 | 4.45 ±0.1 | **4.06**∗ ±**0.1** |
| | | | SVM | 6.11 ±0.1 | 6.29 ±0.1 | 6.41 ±0.1 | **5.48** ±**0.1** |
| | Random Forest | 2.45 ±0.1 | Tree | 4.49 ±0.1 | 4.49 ±0.1 | 4.47 ±0.1 | **3.89** ±**0.1** |
| | | | SVM | 6.11 ±0.1 | 6.16 ±0.1 | 5.96 ±0.1 | **5.53** ±**0.1** |

weight for that input. Note that setting large weights to zero reduces the variance of the simple model because it prevents dependence on a select few examples. Moreover, large weights mostly indicate that the input is extremely hard for the simple model to classify correctly and so expending effort on it and ignoring other examples will most likely be detrimental to performance. Best values for both parameters can be found empirically using standard validation procedures.

## 4 EXPERIMENTS

In this section, we empirically validate our approach as compared with other state-of-the-art methods used to improve simple models. We experiment on 6 real datasets from UCI (Dheeru & Karra Taniski-dou, 2017): Ionosphere, Ovarian Cancer (OC), Heart Disease (HD), Waveform, Human Activity Recognition (HAR), and Musk. Data characteristics are given in the appendix.

### 4.1 UCI DATASETS SETUP

We experiment with two complex models, namely, boosted trees and random forests, each of size 100. For each of the complex models we see how the different methods perform in enhancing two simple models: a single CART decision tree and a linear SVM classifier. Since ProfWeight is not directly applicable in this setting, we compare with its special case ConfWeight which weighs examples based on the confidence score of the complex model. We also compare with two models that serves as a proxy to Distillation, namely `Distill-proxy 1` and `Distill-proxy 2` since distillation is mainly designed for cross-entropy loss with soft targets. For `Distill-proxy 1`, we use the hard targets predicted by the complex models (boosted trees or random forests) as labels for the simple models. For `Distill-proxy-2`, we use regression versions of trees and SVM for the simple models to fit the soft probabilities of the complex models. For multiclass problems, we train a separate regressor for fitting a soft score for each class and choose the class with the largest soft score. This version performed worse and numbers are relegated to the supplement. We only report numbers for `Distill-proxy 1` in the main paper. Datasets are randomly split into 70% train and 30% test. Results for all methods are averaged over 10 random splits and reported in Table 8 with 95% confidence intervals.

For our method, graded classifiers based on the complex models are formed as described before in steps of 10 trees. We have 10 graded classifiers ($10 \times 10 = 100$ trees) for both boosted trees and random forests. The trees in the random forest are ordered based on increasing performance. Optimal values for $\gamma$ and $\beta$ are found using 10-fold cross-validation.

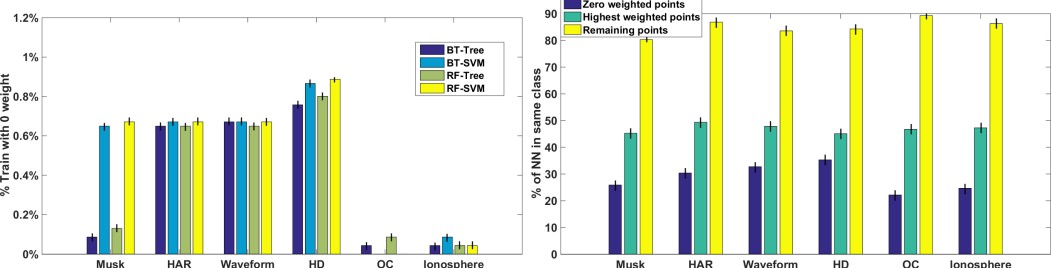

Figure 1: Above (left) we see the % of training set points assigned weight 0 by SRatio at optimal $\beta$ values for each complex (BT, RF) and simple model (Tree, SVM) combination on the 6 UCI datasets. We see that $< 1\%$ of the training set has weights 0 in all cases. Above (right) we analyze why intuitively our reweighting seems to work by considering the % of nearest neighbors that have zero weight, high weight, and in-between weight. Results are averaged over all complex-simple model combinations. Both plots represent averaged values over 10 random train/test splits with 95% confidence intervals.

### 4.2 CIFAR-10 SETUP

The setup we follow here is very similar to previous works (Dhurandhar et al., 2018b). The complex model is an 18 unit ResNet with 15 residual (Res) blocks/units. We consider a simple model that consists of 3 Res units, 5 Res units and 7 Res units. Each unit consists of two $3 \times 3$ convolutional layers with either 64, 128, or 256 filters (the exact architecture is given in the appendix). A $3 \times 3$ convolutional layer with 16 filters serves an input to the first ResNet block, while an average pooling layer followed by a fully connected layer with 10 logits takes as input the output of the final ResNet block for each of the models. [1]

---

[1]Tensorflow 1.5.0 was used for CIFAR-10 experiments

Table 2: Below we observe the averaged accuracies (%) of simple models SM-3 (3 Res units), SM-5 (5 Res units) and SM-7 (7 Res units) trained with various weighting methods and distillation. The complex model achieved $84.5\%$ accuracy. Statistically significant best results are indicated in bold.

|  | SM-3 | SM-5 | SM-7 |
|---|---|---|---|
| Standard | 73.15 ($\pm$ 0.7) | 75.78 ($\pm$0.5) | 78.76 ($\pm$0.35) |
| ConfWeight | 76.27 ($\pm$0.48) | 78.54 ($\pm$0.36) | **81.46** ($\pm$0.50) |
| Distillation | 65.84 ($\pm$0.60) | 70.09 ($\pm$0.19) | 73.4 ($\pm$0.64) |
| ProfWeight | 76.56 ($\pm$0.51) | 79.25 ($\pm$0.36) | **81.34** ($\pm$0.49) |
| SRatio | **77.23** ($\pm$0.14) | **80.14** ($\pm$0.22) | **81.89** ($\pm$0.28) |

We form 18 graded classifiers by training probes which are linear classifiers with softmax activations attached to flattened intermediate representations corresponding to the 18 units of ResNet (15 Res units + 3 others). As done in prior studies, we split the 50000 training samples from the CIFAR-10 dataset into two training sets of 30000 and 20000 samples, which are used to train the complex and simple models, respectively. 500 samples from the CIFAR-10 test set are used for validation and hyperparameter tuning (details in appendix). The remaining 9500 are used to report accuracies of all the models. Distillation (Geoffrey Hinton, 2015) employs cross-entropy loss with soft targets to train the simple model. The soft targets are the softmax outputs of the complex model's last layer rescaled by temperature $t = 0.5$ which was selected based on cross-validation. For ProfWeight, we report results for the area under the curve (AUC) version as it had the best performance in a majority of the cases in the prior study. Details of $\beta$ and $\gamma$ values that we experimented with to obtain the results in Table 2 are in the appendix.

### 4.3 OBSERVATIONS

In the experiments on the 6 UCI datasets depicted in Table 8, we observe that we are consistently the best performer, either tying or superseding other competitors. Given the 24 experiments based on dataset, complex model, and simple model combinations ($6 \times 2 \times 2 = 24$), we are the outright best performer in 14 of those cases, while being tied with one or more other methods for best performance in the remaining 10 cases. In fact, in 2 cases where we are outright best performers, dataset=Ionosphere, complex model =boosted trees, simple model = Tree and dataset=Musk, complex model =boosted trees, simple model = Tree, our method enhances the simple model's performance to match (statistically) that of the complex model. We believe this improvement to be significant. In fact, on the Musk dataset, we observe that the simple tree model enhanced using our method, where the complex model is a random forest, supersedes the performance of the other complex model. On the Ovarian Cancer dataset, linear SVM actually seems to perform best, even better than the complex models. A reason for this may be that the dataset is high dimensional with few examples. Due to this, it also seems difficult for any of the methods to boost the simple model's performance.

We now offer an intuition as to why our weighting works. First, we see in figure 1(left) that our assertion of only a very small fraction of the training set being assigned 0 weights based on parameter $\beta$, which upper bounds the weights, is true ($< 1\%$ is assigned weight 0). For Ovarian Cancer (OC), SVM was better than the complex models and hence no points had weight 0.

In figure 1(right), we see the intuitive justification for the learned weights. Given 10 nearest neighbors (NN) of a data point, let $\nu_s$ and $\nu_d$ denote the number of those NNs that belong to its class (i.e. same class), and most frequent different class respectively. Then the Y-axis depicts $\frac{\nu_s}{\nu_s+\nu_d} \times 100$. This metric gives a feel for how difficult it is likely to be to correctly classify a point. We thus see that most of the 10 NNs for the 0 weight points lie in a different class and so likely are almost impossible for a simple model to classify correctly. The highest weighted points (i.e. top 5 percentile) have nearest neighbors from the same class almost 50% of the time and are close to the most frequent (different) class. This showcases why the 0 weight points are so difficult for the simple models to classify, while the highest weighted points seem to outline an important decision boundary. With some effort a simple model should be able to classify them correctly and so focusing on them is important. The remaining points (on average) seem to be relatively easy for both complex and simple models.

On the CIFAR-10 dataset, we see that our method outperforms other state-of-the-art methods where the simple model has 3 Res units and 5 Res units. For 7 Res units, we tie with ProfWeight and ConfWeight. Given the motivation of resource limited settings where memory constraints can be stringent (Reagen et al., 2016; Chen et al., 2016), SM-3 and SM-5 are anyway the more reasonable options. In general, we see from these experiments that the simple model's predictions can be highly informative in improving its own performance.

## 5 DISCUSSION

Our approach and results outline an interesting strategy, where even in cases that one might want a simple model, it might be beneficial to build an accurate complex model first and use it to enhance the desired simple model. Such is exactly the situation for the manufacturing engineer described in the introduction that has experience with simple interpretable models that provide him with knowledge that a complex model with better performance cannot offer.

Although our method may appear to be simplistic, we believe it to be a conceptual jump. Our method takes into account the difficulty of a sample not just based on the complex model, but also the simple model which a priori is not obvious and hence possibly ignored by previous methods that may or may not be weighting-based. Moreover, we have empirically shown that our method either outperforms or matches the best solutions across a wide array of datasets for different complex model (viz. boosted trees, random forests and ResNets) and simple model (viz. single decision trees, linear SVM and small ResNets) combinations. In fact, in a couple of cases, a single tree approached the performance of a 100 boosted trees using our method. In addition, we also formalized and generalized the idea behind probes presented in previous work (Dhurandhar et al., 2018b) to classifiers beyond deep neural networks and gave examples of practical instantiations. In the future, we would like to uncover more such methods and study their theoretical underpinnings.

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

## A Experimental Details

Table 3: Dataset characteristics, where $N$ denotes dataset size and $d$ is the dimensionality.

| Dataset | $N$ | $d$ | # of Classes |
|---|---|---|---|
| Ionosphere | 351 | 34 | 2 |
| Ovarian Cancer | 216 | 4000 | 2 |
| Heart Disease | 303 | 13 | 2 |
| Waveform | 5000 | 40 | 3 |
| Human Activity | 10299 | 561 | 6 |
| Musk | 6598 | 166 | 2 |
| CIFAR-10 | 60000 | $32 \times 32$ | 10 |

| Units | Description |
|---|---|
| Init-conv | $\begin{bmatrix} 3 \times 3 \text{ conv}, \ 16 \end{bmatrix}$ |
| Resunit:1-0 | $\begin{matrix} 3 \times 3 \text{ conv}, \ 64 \\ 3 \times 3 \text{ conv}, \ 64 \end{matrix}$ |
| (Resunit:1-x)$\times$ 4 | $\begin{matrix} 3 \times 3 \text{ conv}, \ 64 \\ 3 \times 3 \text{ conv}, \ 64 \end{matrix} \times 4$ |
| (Resunit:2-0) | $\begin{matrix} 3 \times 3 \text{ conv}, \ 128 \\ 3 \times 3 \text{ conv}, \ 128 \end{matrix}$ |
| (Resunit:2-x)$\times$ 4 | $\begin{matrix} 3 \times 3 \text{ conv}, \ 128 \\ 3 \times 3 \text{ conv}, \ 128 \end{matrix} \times 4$ |
| (Resunit:3-0) | $\begin{matrix} 3 \times 3 \text{ conv}, \ 256 \\ 3 \times 3 \text{ conv}, \ 256 \end{matrix}$ |
| (Resunit:3-x)$\times$ 4 | $\begin{matrix} 3 \times 3 \text{ conv}, \ 256 \\ 3 \times 3 \text{ conv}, \ 256 \end{matrix} \times 4$ |
| Average Pool | |
| Fully Connected - 10 logits | |

Table 4: 18 unit Complex Model with 15 ResNet units.

Table 5: Residual Network Model used as the complex model for CIFAR-10 experiments in Section 4.2

| Simple Model IDs | Additional Resunits | Rel. Size |
|---|---|---|
| SM-3 | None | $\approx 1/5$ |
| SM-5 | (Resunit:1-x)$\times$1 
 (Resunit:2-x)$\times$1 | $\approx 1/3$ |
| SM-7 | (Resunit:1-x)$\times$2 
 (Resunit:2-x)$\times$1 
 (Resunit:3-x)$\times$1 | $\approx 1/2$ |

Table 6: Additional Resnet units in the Simple Models apart from the commonly shared ones. The last column shows the approximate size of the simple models relative to the complex neural network model in the previous table.

### A.1 Additional Training Details

**CIFAR-10 Experiments**

**Complex Model Training:** We trained with an $\ell$-2 weight decay rate of $0.0002$, sgd optimizer with Nesterov momentum (whose parameter is set to $0.9$), 600 epochs and batch size 128. Learning rates are according to the following schedule: $0.1$ till $40k$ training steps, $0.01$ between $40k$-$60k$

| Probes | 1 | 2 | 3 | 4 | 5 | 6 | 7 | 8 | 9 |
|---|---|---|---|---|---|---|---|---|---|
| Training Set 2 | 0.298 | 0.439 | 0.4955 | 0.53855 | 0.5515 | 0.5632 | 0.597 | 0.6173 | 0.6418 |
| Probes | 10 | 11 | 12 | 13 | 14 | 15 | 16 | 17 | 18 |
| Training Set 2 | 0.66104 | 0.6788 | 0.70855 | 0.7614 | 0.7963 | 0.82015 | 0.8259 | 0.84214 | 0.845 |

Table 7: Probes at various units and their accuracies on the training set 2 for the CIFAR-10 experiment. This is used in the $\mathrm{ProfWeight}$ algorithm to choose the unit above which confidence scores needs to be averaged.

training steps, $0.001$ between $60k - 80k$ training steps and $0.0001$ for $> 80k$ training steps. This is the standard schedule followed in the code by the Tensorflow authors[2]. We keep the learning rate schedule invariant across all our results.

**Simple Models Training:**

1. **Standard**: We train a simple model as is on the training set 2.

2. **ConfWeight**: We weight each sample in training set 2 by the confidence score of the last layer of the complex model on the true label. As mentioned before, this is a special case of our method, ProfWeight.

3. **Distilled-temp-$t$**: We train the simple model using a cross-entropy loss with soft targets. Soft targets are obtained from the softmax ouputs of the last layer of the complex model (or equivalently the last linear probe) rescaled by temperature $t$ as in distillation of Geoffrey Hinton (2015). By using cross validation, we pick two temperatures that are competitive on the validation set ($t = 0.5$ and $t = 40.5$) in terms of validation accuracy for the simple models. We cross-validated over temperatures from the set $\{0.5, 3, 10.5, 20.5, 30.5, 40.5, 50\}$.

4. **ProfWeight** ($>= \ell$): Implementation of our $\mathrm{ProfWeight}$ algorithm where the weight of every sample in training set 2 is set to the averaged probe confidence scores of the true label of the probes corresponding to units above the $\ell$-th unit. We set $\ell = 13, 14$ and $15$. The rationale is that unweighted test scores of all the simple models in Table 2 are all below the probe precision of layer 16 on training set 2 but always above the probe precision at layer 12. The unweighted (i.e. Standard model) test accuracies from Table 2 can be checked against the accuracies of different probes on training set 2 given in Table 5 in the supplementary material.

5. **SRatio**: We average confidence scores from $\ell = 13, 14$ and $15$ as done above for ProfWeight and divide by the simple models confidence. In each case, we optimize over $\beta$ which is increased in steps of $0.5$ from $1.5$ to $10$.

## A.2   EXPERIMENTAL RESULTS FOR DISTILL-PROXY 2

We provide results for the second variant of Distillation `Distill-proxy 2` in Table 8.

---

[2]Code is taken from:
https://github.com/tensorflow/models/tree/master/research/resnet.

Table 8: Below we see the averaged % errors with 95% confidence intervals for Distill-proxy 2 (regression versions of trees and SVM for the simple models that fit soft probabilities from the complex models) on the six real datasets. The results reported using Distill-proxy 1 are in the main paper and are superior to these. Boosted Trees and Random Forest (100 trees) are the complex models (CM), while a single decision tree and linear SVM are the simple models (SM).

| Dataset | Complex Model | CM Error | Simple Model | SM Error | Distill-proxy 2 Error (SM) |
|---|---|---|---|---|---|
| Ionosphere | Boosted Trees | 8.10 ±0.4 | Tree | 10.95 ±0.4 | 10.95 ±0.4 |
| | | | SVM | 12.38 ±0.6 | 12.17 ±0.3 |
| | Random Forest | 6.19 ±0.4 | Tree | 10.95 ±0.4 | 10.95 ±0.4 |
| | | | SVM | 12.38 ±0.6 | 12.38 ±0.6 |
| Ovarian Cancer | Boosted Trees | 4.68 ±0.4 | Tree | 15.62 ±0.8 | 15.62 ±0.8 |
| | | | SVM | 1.56 ±0.4 | 1.56 ±0.4 |
| | Random Forest | 6.25 ±0.8 | Tree | 15.62 ±0.8 | 15.62 ±0.8 |
| | | | SVM | 1.56 ±0.4 | 1.56 ±0.4 |
| Heart Disease | Boosted Trees | 15.55 ±0.6 | Tree | 23.88 ±0.7 | 23.69 ±0.2 |
| | | | SVM | 17.22 ±0.2 | 17.01 ±0.1 |
| | Random Forest | 15.88 ±0.6 | Tree | 23.88 ±0.7 | 23.88 ±0.7 |
| | | | SVM | 17.22 ±0.2 | 17.22 ±0.2 |
| Waveform | Boosted Trees | 12.96 ±0.1 | Tree | 25.43 ±0.2 | 25.26 ±0.1 |
| | | | SVM | 14.70 ±0.2 | 15.39 ±0.1 |
| | Random Forest | 10.90 ±0.1 | Tree | 25.43 ±0.2 | 25.43 ±0.2 |
| | | | SVM | 14.70 ±0.2 | 14.54 ±0.0 |
| Human Activity Recognition | Boosted Trees | 6.32 ±0.0 | Tree | 7.93 ±0.2 | 7.93 ±0.1 |
| | | | SVM | 14.56 ±0.1 | 16.04 ±0.2 |
| | Random Forest | 2.34 ±0.0 | Tree | 7.93 ±0.2 | 7.45 ±0.1 |
| | | | SVM | 14.56 ±0.1 | 14.23 ±0.3 |
| Musk | Boosted Trees | 4.06 ±0.1 | Tree | 4.49 ±0.1 | 6.11 ±0.1 |
| | | | SVM | 6.11 ±0.1 | 6.34 ±0.1 |
| | Random Forest | 2.45 ±0.1 | Tree | 4.49 ±0.1 | 4.49 ±0.1 |
| | | | SVM | 6.11 ±0.1 | 6.19 ±0.2 |

