# OpenReview forum: "Leveraging Simple Model Predictions for Enhancing its Performance"
_ICLR.cc/2020/Conference — Reject_

### Official Review · AnonReviewer1 · 2019-10-24
**Official Blind Review #1**

**Rating:** 6

**Review:**

The paper proposes a means of improving the predictions of a "simple" (low-capacity) model. The idea is to take the predictions of a "complex" (high-capacity) model, and weight the loss in the simple model based on the ratio of complex to simple models' predictions. Intuitively, this seeks to focus on instances which the complex model can fit, but the unweighted simple model cannot. Experiments show the proposed method to have some benefits over existing approaches.

The application of importance-weighting to the "model distillation" problem is interesting, and the paper gives a reasonable intuition for why this approach might work. One general comment is that in contrasting their approach to a number of existing approaches, they note that several of them are typically employed with fairly complex simple models (e.g., neural networks). This may be true, but it was not clear that any of them require this to be the case. Surely they can also be used with simple models as ones you consider in the paper? In this case, I would've liked more elucidation as to why the proposed method can be expected to offer superior performance.

The theoretical justification of the approach is provided by means of Lemmas 3.1 and 3.2, for which I have some comments:
- Lemma 3.1: the notation here is a bit imprecise. In general, a loss for example (x, y) takes in a true label y and predicted score z(x). You refer to the loss of a probabilistic prediction p(y | x), but do not refer to the actual label y itself. From the proof, it is implicit that you are considering y to be binary, and the use of a margin loss. This is ok, but for the hinge loss one doesn't use a probability estimate p(y | x) as input to the loss, but rather, a real-valued score.

I think the result itself could be proven by noting that if φ is non-increasing and q < p, then φ(p)/φ(q) <= 1 < p/q.

- Lemma 3.2: the result is interesting, but it seems that for practical purposes you are only using the first term, since it is the only quantity that depends on θ. Since max(1, .) >= 1 and -log pθ(y | x) >= 0, it seems one could trivially bound the LHS by the first term since -log pθ(y | x) <= max(1, .) * -log pθ(y | x)? Would this not suffice for the purposes of justifying your method? It should also be noted here how the subsequent requirement that the weights be capped (so as to prevent outliers) fits into the analysis.

In describing the method itself, the authors introduce a notion of δ-graded subsets. I found this to be a bit difficult to parse, and it was not clear why this notion was needed. It does not seem to feature in the description of Algorithm 1, nor the subsequent discussion. On the other hand, I felt that the meaning and need for parameters β and γ ought to have been discussed more prominently upfront.

The experiments show favourable performance of the proposed method over baselines, including the distillation approach of Hinton. The datasets are mostly small-scale, but this is in keeping with the goal of the paper, viz. addressing scenarios where simple models may be desired. Per earlier comments, I did not have a deep sense of what additional information the proposed method exploits so as to improve performance. I gather that the weighting including the predictions of the simple model is one difference; it might have been nice to give a sense of what fraction of points this amplifies or suppresses, compared to just using the predictions of the complex model.

There is a nice illustration in Fig 1 (right) as to the class-labels of the training samples assigned various weights. This shows that points with low weight tend to have low agreement with their neighbours' labels. One question is how using a nearest neighbour probability estimate itself (i.e., using this as the "complex" model) would fare.

Minor comments:
- the title is a bit confusing. It is not clear what "its" refers to. You are leveraging the predictions of a complex model to enhance those of a simple model?
- the text has a number of long sentences that could benefit from rewriting or splitting.
- proof of Lemma 3.2, use \cdot not *.
- proof of Lemma 3.2, θ* should use superscript.
- Fig 1, the caption is overlong. Most of this should be in the text.
- Fig 1, use crisper fonts for the text.

**Experience Assessment:**

I have read many papers in this area.

**Review Assessment: Checking Correctness Of Derivations And Theory:**

I assessed the sensibility of the derivations and theory.

**Review Assessment: Checking Correctness Of Experiments:**

I assessed the sensibility of the experiments.

**Review Assessment: Thoroughness In Paper Reading:**

I read the paper at least twice and used my best judgement in assessing the paper.

---

> ### Author Response · Authors · 2019-11-08
> **Paper Revised - Lemma 3.2 modified (as Lemma 3.1 now) to include the clip level.**
>
> Thank you very much for the comments.  We below address them and have posted a new version of the paper.
>
> Regarding Lemma 3.2: You are correct about the trivial bound, but the bound would not be as tight without the other term. In any case, please refer to our response about Lemma 3.2 to Reviewer 4. We have modified the result to include the requirement that weights be capped, which also provides further intuition into the algorithm and how to select the cap.
>
> Regarding comparison with existing methods: We added a paragraph discussing relationships between our work and the existing ones that Reviewer 4 pointed out in the related work.
>
> Regarding delta-graded classifiers: Again, please refer to our response to Reviewer 4. The notion of delta-graded classifiers is necessary in order to extract information from a a large class of models and generalizes how Dhurandhar et al (2018) attach probes to different layers of a neural network to generate predictions from various layers of a neural network. As in Dhurandhar et al (2018), using such information proved to be useful here, in comparison to simply using the final prediction of a model (which is labelled ConfWeight in our experiments). We further clarify this in the beginning of Section 3 (Methodology).
>
> Regarding what additional information the proposed method exploits: Indeed, using a weighting that includes the predictions of the simple model is the key difference from all other works, and judging from our experiments, is a significant differentiator from other weighting schemes.
>
> Regarding what fractions of points were suppressed: Figure 1 (left) demonstrates that less than 1% of points were suppressed as being too difficult for the simple model (no points are suppressed by only using the complex model predictions).
>
> Regarding Minor Comments: Please note that, regarding the title, we are leveraging the simple model predictions (along with the complex model predictions) to enhance those of a simple model, so "its" refers to the Simple model. We can remove "its" if it is still not clear. We have made a thorough pass and did some rewriting, fixing long sentences, and improving the overall flow. Most of the caption to Figure 1 has been moved to the text.

---

### Official Review · AnonReviewer2 · 2019-10-30
**Official Blind Review #2**

**Rating:** 6

**Review:**

This paper is about re-weighting training sets in such a way that simple and interpretable models like trees or small networks can mimic the performance of potentially very complex learning architectures. The main difference to some existing approaches of this kind is the universal applicability to many complex-simple model combinations (and not only layered networks). Technically, this paper is an extension and formalization of some ideas proposed in (Dhurandhar 2018). The main methodological contribution is the formal justification of the proposed weighting scheme for samples which utilizes the ratio of conditional probabilities p_complex(y|x)/p_simple(y|x) in the Lemma 3.1 and 3.2. The practical algorithm described in the paper is based on the concept of delta-graded classifiers which basically defines a nested set of classifiers of increasing accuracy -- such as, for instance, simple classifiers trained on intermediate layers of a deep network. The second important input argument is the learning algorithm for the simple model, which is trained on the re-weighted samples. The motivation for this algorithmic procedure is somewhat "hand-waving", but rather intuitive. Experiments for different benchmark datasets and different complex-simple model combinations nicely demonstrate that this method is indeed quite useful in practice. In summary, I think that this work addresses a highly relevant problem and provides a relatively simple method that might be very useful in many applications. A potential weakness could be the gap between the formal treatment of the cross-entropy loss in Lemma 3.2. and the rather "ad hoc" practical algorithm that crucially depends on two tuning parameters. Nevertheless, in my opinion, this work provides some interesting ideas that have the potential to trigger future research in this area.

**Experience Assessment:**

I have published in this field for several years.

**Review Assessment: Checking Correctness Of Derivations And Theory:**

I assessed the sensibility of the derivations and theory.

**Review Assessment: Checking Correctness Of Experiments:**

I carefully checked the experiments.

**Review Assessment: Thoroughness In Paper Reading:**

I read the paper at least twice and used my best judgement in assessing the paper.

---

> ### Author Response · Authors · 2019-11-08
> **Paper revised - Lemma 3.2 changed to reflect the clip level of the weights**
>
> Thank you very much for your feedback. Please refer to our response to Reviewer 4 regarding Lemma 3.2. We have modified the Lemma in the revision to include the clip parameter beta, and the motivation for the algorithm now follows directly, with new intuitions. Gamma must still be tuned.

---

### Official Review · AnonReviewer3 · 2019-11-01
**Official Blind Review #3**

**Rating:** 6

**Review:**

This paper presents a simple yet effective trick to reweigh the samples fed into a simple model to improve the performance. The new weights are functions of both the simple model and another complex (and strong) model. Some theoretical and intuitive explanations are provided to support the main claim of the paper.
The main weakness of the paper is its presentation and this is apparent from the abstract of the paper and afterwards. Referring to a prior work (Dhuranhar, 2018b) in the abstract which might be known to the reader is not a good idea. The connection of lemma 3.1 to the rest of the paper is not clear. Similarly, it's not readily clear how lemma 3.2 leads to algorithm 1. The introduction of the graded classifiers was very sudden. It's difficult to pinpoint how we replaced the complex model with the graded classifiers. To me, the graded classifiers were and incremental work and make the main point of paper (section 3.1 and 3.2) could have been emphasized better.
What would be the effect of changing the number of graded classifiers? Some theoretical explanations or empirical results could be beneficial.
The reweighting scheme looks very related to importance sampling ratios in off policy evaluation in reinforcement learning and counterfactual analysis. Connecting the dots could be interesting.

**Experience Assessment:**

I do not know much about this area.

**Review Assessment: Checking Correctness Of Derivations And Theory:**

I assessed the sensibility of the derivations and theory.

**Review Assessment: Checking Correctness Of Experiments:**

I assessed the sensibility of the experiments.

**Review Assessment: Thoroughness In Paper Reading:**

I read the paper at least twice and used my best judgement in assessing the paper.

---

> ### Author Response · Authors · 2019-11-08
> **Paper revised - clarified what graded classifiers are and Lemma 3.2 modified and changed.**
>
> Thank you for your comments. We hope to address your concerns with the following responses:
>
> Regarding the presentation of the work and graded classifiers: We have made a thorough pass through the paper to address the writing, as well as take into account your specific comments such as our introduction to the graded classifiers (please refer to the beginning of Section 3 Methodology).  Graded classifiers are a generalization of the concept of probes used for neural networks in Dhurandhar et al 2018. In short, the graded classifiers extract useful intermediate information from the complex classifiers, in the same way that a probe could be attached to an intermediate layer of a neural network to see how useful the layers up to a certain point are for prediction. The parameter gamma could be chosen such that I only includes the final prediction of the complex model, however experiments have shown that using the mean of predictions from other models in the set of what we call delta-graded classifiers gives better performance in our framework than using only the final complex model. We have put more emphasis on Sections 3.1 and 3.2, which we think will be greatly improved by our modified Lemma 3.2.
>
> Regarding the effect of changing the number of graded classifiers: The number of classifiers to use is tuned by paramater gamma. For the case of neural networks, Dhurandhar et al 2018 already showed that it was useful to use more than only the final output of the complex model, as well as to throw out graded classifiers that achieved worse performance than the simple model (albeit using weights that were not divided by the simple model predictions as we do here).
>
> Regarding Lemma 3.2's connection to algorithm 1: Please see the response to Reviewer 4. We have modified the result to include the requirement that weights be capped, which also provides further intuition into the algorithm and how to select the cap.
>
> Regarding connections to importance sampling ratios in off policy evaluation in reinforcement learning and counterfactual analysis: Indeed, our scheme looks very related and we have tried to make such connections but have not been able to do so as of yet.

---

### Official Review · AnonReviewer4 · 2019-11-04
**Official Blind Review #4**

**Rating:** 1

**Review:**

The paper proposes a method for transferring knowledge from a complex, high-performing model to a small/lower-performing one. The approach is based on reweighting the train examples according to the ratio of confidences of the complex and simple models, and then retraining the simple model on the reweighed dataset.

Pros:
* Some interesting ideas motivating the approach

Cons:
* An important error in the main result and various other issues (imprecise statements, lack of details) raise doubts on the theoretical contribution of the paper.
* Limited novelty. Contribution boils down to importance reweighting via a ratio of probabilities.
* Limited awareness of related work.
* Experimental framework lacks meaningful baselines and important details .

Based on the points above, I have to recommend rejection of this paper to ICLR.

Detailed Comments:
* The claim that per-sample hardness reweighting is novel to this work is a bit of a stretch. Most reweighting-based methods for distillation use some form of this idea, though how each method estimates “hardness” varies, ranging from raw confidence scores of the teacher model (e.g., classic distillation), to scaled versions of it, to a ratio of this and the student’s score (this work).
* More generally, there’s quite a few very relevant related works which are not cited here (see list below for a non-exhaustive list).
* I believe the inequality used to prove Lemma 3.2 is not true in general. It’s easy to come up with counterexamples for which w>=1 and x>1 yet log(wx) <= w log(x) does not hold. For example, for w=2, any x in (1,2) will not satisfy this. In general, this inequality is only true if x >= w^{1/(w-1)}. I would be prepared to pass this off as a minor error, were it not for the fact that the w and x in the proof of Lemma 3.2 could quite conceivably be outside this range, i.e., rendering the inequality (and therefore the bound) invalid.
* There are various other technical aspects of the paper that are either non-rigorous or lack details. For example, Lemma 3.1 requires more information. For which y does w>= hold? Is it just for the true class y*? The proof is equally obscure, and seems to use the same problematic inequality of Lemma 3.2 (for the cross-entropy loss).
* The statement of Lemma 3.2 is confusing. The last sentence should read loss smaller than *that* of theta*. Also, I don’t see the optimality of theta* being used anywhere. The statement could very well be phrased in terms of any two models, one of which has a lower loss than the other.
* Assuming the complex “teacher” classifiers are very accurate (which is assumed several times throughout the paper), re-weighting training examples by max(1, p_c / p_theta) does not provide too much information beyond the ground truth labels. This is a core problem in knowledge distillation, already identified as early as Hinton et al (2015), and which is often alleviated by some form of temperature annealing or by operating on pre-softmax logits rather than final probabilities. That doesn’t seem to be the case here, so I’m quite puzzled by the “intuitive justification” provided in this work.
* The discussion in the first paragraph of pg 4 is a well known dichotomy (namely, up-weighting easy or hard examples) in the distillation/boosting/instance-reweighting literature. This discussion would really benefit from awareness of previous work discussing this duality (e.g., (Bengio et al 2009; Ren et al 2019)
* Where and how is the ordering of the “graded” classifiers used? In step (3) of Algo 1 all \zeta’s seem to be weighted equally, so I’m missing why the classifiers need to be delta-graded.
* I find it quite surprising that in almost half of the datasets in Table 1, distillation yields the exact same performance as just training the Simple Model, and sometimes even degrades the performance. Section 4.1 points out that this is actually “an equivalent model to Distillation”, but no further explanation is given, so it is hard to conjecture what might be the problem. Futhermore, as the authors point out in the introduction, Distillation alla Hinton 2015 usually assumes both teacher and student models are soft predictors (e.g., neural nets), but the models used in the UCI experiments trees and SVMs, so I wonder how exactly they did this.
* For problems involving distillation into simple interpretable models (like trees), such as the UCI experiments, there are many reasonably strong baselines from previous work that could have been compared against (e.g. Frosst and Hinton 2017), which could have provided a more meaningful evaluation.


[1] Learning to Reweight Examples for Robust Deep Learning, Ren et al. ICML 2018
[2] Distilling a Neural Network Into a Soft Decision Tree, Frosst and Hinton, 2017
[3] Born-Again Neural Networks, Furlanello et al., ICML 2018
[4] Fidelity-Weighted Learning, Deghani et al., ICLR 2018

**Experience Assessment:**

I have published one or two papers in this area.

**Review Assessment: Checking Correctness Of Derivations And Theory:**

I carefully checked the derivations and theory.

**Review Assessment: Checking Correctness Of Experiments:**

I assessed the sensibility of the experiments.

**Review Assessment: Thoroughness In Paper Reading:**

I read the paper thoroughly.

---

> ### Author Response · Authors · 2019-11-08
> **Paper revised based on your comments - Explanation of distillation methods used included and a Modified Lemma 3.2 added**
>
> Thank you very much for the detailed comments and thorough review. We have addressed most of your comments below and posted a new version of the paper.
>
> Regarding Lemma 3.1: The inequality we use for the cross-entropy loss is actually correct - note that it holds for $0 \leq p,q \leq 1$ where the weight is $p/q$ (for Lemma 3.2, the weight is a function of a different pretrained model's q so the needed inequality is different). The inequality holds for the true class y* throughout Lemma 3.1. Regardless, we do agree that the connection to Algorithm 1 is not direct as it is for Lemma 3.2 (see below) as other reviewers note, so we have removed Lemma 3.1 to improve the flow of the paper.
>
> Regarding Lemma 3.2: Thank you for finding the mistake. In fact, if we assume that the weights are clipped at a level beta (which we do in our experiments), then one can fix the inequality by introducing a slack term which is logarithmic in beta, i.e., $\log(wx) \leq w \log(x) + \log(\beta)$ for all $\beta \geq w \geq 1$ and $x > 1$. This leads to a bound where the slack term is log(beta) - E[log of the clipped weight] which is always positive and characterizes the bias due to clipping. This offers users a way to tune the bound.
>
> Regarding $\theta*$: You are right that we need not consider the optimal simple model. We mean theta* to be the result of any simple model's training algorithm, without any weighting. It is indeed important that $p_c/p_{\theta*}$ stays above 1 with high probability so that the slack term remains small. This motivates using a complex model for p_c.
>
> Regarding your comment "Assuming the complex teacher classifiers are very accurate ... reweighting training examples by $\max(1, p_c/p_{\theta*})$ does not provide too much information...": This suggestions seems true if our weight was simply $p_c$, however we're using $p_c/p_{\theta*}$, and this weight can be arbitrarily larger than 1. Consider the case when the complex model is much more confident than the simple model. Inclusion of $p_{\theta*}$ into this weight is the key novelty in this work. To our knowledge, no other work uses information from the simple model in a transfer framework. This will lead to a higher weighting of the point, whereas distillation tries to fit to a soft label $p_c$ (possibly temperature rescaled) which is a different transfer framework.
>
> Regarding per-sample hardness reweighting: We do not claim that per-sample hardness reweighting is novel. We clearly refer to other works that have done such schemes (e.g. Dhurandhar et al 2018). Our novelty claim is specific to the use of a ratio that makes use of the simple model, which was not done before. This use of simple model scores for the weights is not obvious apriori and we utilize this and show improvements. Further, we would like to clarify that classic distillation is not a reweighting scheme in the sense that it does not optimize a loss of the form $\sum_i w_i f(x_i, y_i)$. Instead, it actually fits a simple model to a soft label as described above. Suppose the soft targets are obtained from a deep neural net, the soft label function of x will have exponentially many linear pieces in the depth of the network. Temperature scaling does not affect this complexity of the soft label function. One can fit to this if the simple model also permits that many linear pieces (or has sufficient complexity). But our method is just sample reweighting and does not attempt to fit a high dimensional complex function.
>
> Regarding the dichotomy in up-weighting easy and hard examples. Indeed, Bengio 2009 addresses such a dichotomy in that the ordering of such examples during training can achieve better learning outcomes. Typically, this ordering is done manually for their method. The key there is to compare easy versus hard examples; if we knew which examples were easy versus hard, it would be better to train on the easy examples first followed by the hard ones. However, we offer a weighting that says examples considered easy by the complex model and hard by the simple should be given priority through weighting (no ordering of samples) in training the simple model. The question of easy versus hard is considered for each example here (rather than to order examples) in that we care about the ratio of the hardness between the complex and simple model.
>
> Regarding graded classifiers: Graded classifiers here generalize the work of Dhurandhar et al 2018 for Neural Networks to other models. In Step (3) of Algo 1, all $\zeta$'s in the set I are weighted equally (as in previous work). However, the set I removes $\zeta$'s that are not within \gamma accuracy of the simple model. The ordering (based on delta-graded) is important in order to obtain information from models of progressive accuracy (that are not much worse than the accuracy of the simple model without reweighting). The delta is important so that the ordering holds true for a part of the domain with sufficient probability mass (delta).

---

> > ### Author Response · Authors · 2019-11-08
> > **Previous Rebuttal to Reviewer 4 continued due to character limits**
> >
> > Regarding our experiments with distillation for SVM and trees: We tried two versions which we will refer to as distillation proc-1 and distillation proc-2 and change accordingly in the paper. 1) We use the hard targets predicted by the complex models as outputs for the simple models. 2) We use the regression versions of tree and svm and predict the soft probabilities of the complex models. In the case of multi class datasets, we train separate regressors for each label and choose the label with the best soft score for a point on the test. The first scheme gave better results so we reported those in the paper and  wrote this was an equivalent model to distillation. We will change this and make clear what that means here.
> >
> > Regarding the related works: Thank you for pointing them out. We have cited and discuss them in the Section 2 of the revised paper, however we do not view them as baselines for our current work. [1] is about learning weights while training a model, where the weights are a tuned on a validation set with the goal of robustifying the model, which is a very different goal. There is no transfer between models, nor is there a goal to improve performance of the pretrained model. [2] distills information specifically from a neural network to a soft decision tree, where a soft decision tree is a new model class that they have derived, and again, fits only to soft labels. It is applicable only for this specific scenario. [3] distills information from one model to another model of equivalent complexity and fits to the first model's soft scores, again however there is no student model of inferior performance as in our framework. [3]'s use of a method the call Confidence-Weighted by Teacher Max is equivalent to confWeight (from Dhurandhar et al 2018) which we include in our experiments. [4] uses a Gaussian Process regression to estimate uncertainty in the soft scores, and then during student training using stochastic gradient descent, they use these uncertainty estimates to change the stepsize. This only works if SGD is used for training the simple model (not assumed in our work) and further we do not have access to uncertainty of the soft estimates of the complex model. They have a small set of strongly annotated labels over which the GP is trained to provide soft scores and its variance. This is then used by the student to learn weakly annotated data. It is a very different problem.

---

### Decision · Program_Chairs · 2019-12-19

**Decision:**

Reject

**Comment:**

The authors propose a sample reweighting scheme that helps to learn a simple model with similar performance as a more complex one. The authors contained critical errors in their original submission and the paper seems to lack in terms of originality and novelty of the proposed method.